# SPATIAL-WINOGRAD PRUNING ENABLING SPARSE WINOGRAD CONVOLUTION

## ABSTRACT

Deep convolutional neural networks (CNNs) are deployed in various applications but demand immense computational requirements. Pruning techniques and Winograd convolution are two typical methods to reduce the CNN computation. However, they cannot be directly combined because Winograd transformation fills in the sparsity resulting from pruning. Li et al. (2017) propose sparse Winograd convolution in which weights are directly pruned in the Winograd domain, but this technique is not very practical because Winograd-domain retraining requires low learning rates and hence significantly longer training time. Besides, Liu et al. (2018) move the ReLU function into the Winograd domain, which can help increase the weight sparsity but requires changes in the network structure. To achieve a high Winograd-domain weight sparsity without changing network structures, we propose a new pruning method, *spatial-Winograd pruning*. As the first step, spatial-domain weights are pruned in a structured way, which efficiently transfers the spatial-domain sparsity into the Winograd domain and avoids Winograd-domain retraining. For the next step, we also perform pruning and retraining directly in the Winograd domain but propose to use an importance factor matrix to adjust weight importance and weight gradients. This adjustment makes it possible to effectively retrain the pruned Winograd-domain network without changing the network structure. For the three models on the datasets of CIFAR-10, CIFAR-100, and ImageNet, our proposed method can achieve the Winograd-domain sparsities of 63%, 50%, and 74%, respectively.

## 1 INTRODUCTION

Deep convolutional neural networks (CNNs) have been ubiquitously utilized in various application domains. However, their performance comes at the cost of a significant amount of computation which keeps growing over time. As an example, for the ImageNet challenge (Russakovsky et al., 2015), Krizhevsky et al. (2012) proposed AlexNet which requires more than $1.1 \times 10^9$ multiplications. Later, in 2016, the ResNet-152 model (He et al., 2016) increased the computation cost to $11.3 \times 10^9$ multiplications. This high computation cost limits the deployment of larger and deeper CNN models.

There are two primary methods to reduce the required computation of CNN models: pruning techniques and Winograd/FFT convolution. Pruning removes redundant weight parameters, inducing sparsity into the network. On the other hand, Winograd convolution (Lavin & Gray, 2016) and FFT convolution (Mathieu et al., 2013) transform the computation into different domains. The convolution operations can then be replaced by element-wise multiplications. For the typical convolution kernel size of $3 \times 3$, Winograd convolution can achieve more than twofold speedup over highly optimized spatial convolution algorithms, and typically requires fewer flops than FFT-based approaches (Li et al., 2017). Therefore, in this paper, we focus on the Winograd convolution.

The pruning techniques and Winograd convolution are not directly compatible with each other. Sparse weight matrices, which are generated by pruning, lose most of the sparsity after the Winograd transformation from the spatial (original) domain to the Winograd domain. The remaining sparsity is much lower than what we need for improving computation performance.

To increase the Winograd-domain sparsity, Li et al. (2017) propose to perform pruning and retraining directly on Winograd-domain weights. However, it requires using an extremely small learning rate,

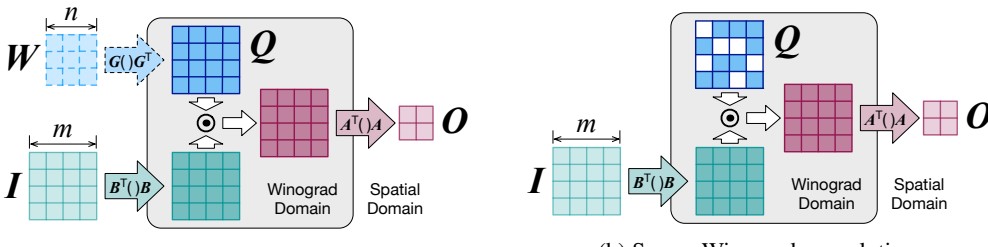

(a) Conventional Winograd convolution.          (b) Sparse Winograd convolution.

Figure 1: Conventional Winograd convolution and sparse Winograd convolution ($m = 4$, $n = 3$).

e.g., 200x smaller for AlexNet, in retraining and is difficult to be applied to deep networks. Besides, Winograd-ReLU pruning (Liu et al., 2018) moves ReLU function into the Winograd domain, which helps increase Winograd-domain sparsity but requires changes in the network structure.

In this paper, to further improve the sparsity of Winograd-domain weights without changing the network structure, we propose a new pruning method, *spatial-Winograd pruning*. It includes two parts: *spatial structured pruning and Winograd direct pruning*. In spatial structured pruning, we prune the spatial-domain weights in a structured way, in which the structures are designed to transfer the spatial-domain sparsity into the Winograd domain efficiently. After spatial structured pruning, weights of the pruned layers will be converted to and kept in the Winograd domain. Then, for Winograd direct pruning, we perform pruning and retraining entirely in the Winograd domain to improve the sparsity further.

This paper makes the following contributions:

- We propose a new pruning method, *spatial-Winograd pruning*. Without changing the network structure, it can achieve higher sparsity in Winograd-domain weights compared with previous methods.

- As the first part of spatial-Winograd pruning, we provide a structured pruning method to transfer the spatial-domain sparsity into the Winograd domain efficiently. It can help avoid Winograd-domain retraining in this part and accelerate the pruning process.

- In the second part, to perform pruning directly in the Winograd domain, we present a new approach to measuring the importance of each Winograd-domain weight based on its impact on output activations. Also, we propose to use an importance factor matrix to adjust the gradients of Winograd-domain weights, which makes it much faster to retrain deep networks directly in the Winograd domain without changing the network structure.

## 2 PRELIMINARY AND RELATED WORK

Winograd convolution (Lavin & Gray, 2016) is a typical algorithm to reduce the arithmetic complexity of CNNs. It transforms the computation into the Winograd domain, and the convolution operations can then be replaced by element-wise multiplications. We call the domain, in which the conventional convolution operation is executed, to be the spatial domain.

The basic block of Winograd convolution works on a 2D input tile, $I$, with a size of $m \times m$ and a 2D weight filter, $W$, with a size of $n \times n$. In this case, the 2D output tile generated, $O$, will have a size of $(m - n + 1) \times (m - n + 1)$. For a typical convolutional layer, the input feature maps are first disassembled into input tiles and, after the Winograd convolution, the output tiles will be reassembled into the output feature maps.

Figure 1a shows how conventional Winograd convolution works. As the first step, the weight filter $W$ and the input tile $I$ are converted into the Winograd domain using the predefined matrices $G$ and $B$. Element-wise multiplication is then applied to the Winograd-domain weight filter, $GWG^\top$, and input tile, $B^\top I B$, to generate the Winograd-domain output tile with a size of $m \times m$. In the last step, the output tile is converted back into the spatial domain with another predefined matrix $A$. With $\odot$ as the Hadamard product (element-wise multiplication), the entire process can be written as

$$O = A^\top[(GWG^\top) \odot (B^\top I B)]A \qquad (1)$$

Figure 2: Overview of the spatial-Winograd pruning.

The transform/inverse-transform matrices $\boldsymbol{A}$, $\boldsymbol{B}$ and $\boldsymbol{G}$ are only determined by $m$ and $n$. These matrices contain many repeating elements and applying them requires only few multiplications. In this case, considering only the element-wise multiplication between $\boldsymbol{GWG}^\top$ and $\boldsymbol{B}^\top \boldsymbol{I}\ \boldsymbol{B}$, the Winograd convolution can reduce the number of multiplications from $(m-n+1)^2 n^2$ to $m^2$.

In addition to Winograd convolution, pruning is also a well-explored method to reduce CNN computation. Han et al. (2015b;a) propose to perform pruning and retraining iteratively, which can help reduce the computation by up to $5\times$. To fully utilize the sparsity incurred by pruning to accelerate CNN computation, Wen et al. (2016) and Yu et al. (2017) prune networks in a structured way: weights are clustered into groups with hardware-friendly structures and then get pruned in groups.

However, Winograd convolution is not directly compatible with conventional pruning algorithms. The transformation $\boldsymbol{GWG}^\top$ fills in the zeros in the sparse weight filters generated by pruning. There have been several research attempts to solve this problem.

Liu & Turakhia (2016) propose to directly mask out Winograd-domain weights and use backpropagation to train the spatial-domain weights. However, compared with spatial-domain weights, Winograd-domain weights are in a higher-dimensional space. Directly setting Winograd-domain weights to zero will cause an inconsistency between the spatial domain and the Winograd domain. This inconsistency will lead to a significant accuracy loss or a low sparsity on networks, e.g., AlexNet, for large datasets (Li et al., 2017).

To address the inconsistency between the spatial and Winograd domain, Li et al. (2017) propose the sparse Winograd convolution. Figure. 1b shows how it works. Weight values are stored in the Winograd domain instead of the spatial domain. Both pruning and retraining are applied directly to Winograd-domain weights. This native pruning algorithm achieves $> 90\%$ sparsity on AlexNet (Krizhevsky et al., 2012) but cannot provide a high sparsity for deep networks (Liu et al., 2018). Also, direct retraining in the Winograd domain requires an extremely small learning rate, e.g., 200x smaller for AlexNet, which makes the retraining much slower.

Based on sparse Winograd convolution, Liu et al. (2018) introduce the Winograd-ReLU pruning. It moves the ReLU function from the spatial domain into the Winograd domain. In this case, the computation of Winograd convolution becomes

$$\boldsymbol{O} = \boldsymbol{A}^\top[(\boldsymbol{GWG}^\top) \odot \mathrm{ReLU}(\boldsymbol{B}^\top \boldsymbol{I}\ \boldsymbol{B})]\boldsymbol{A} \qquad (2)$$

The Winograd-domain inputs also become sparse, which helps further reduce the required computation. Besides, a higher sparsity in Winograd-domain weights can be achieved. However, with weight filters being sparse, it is challenging to utilize both the weight and input sparsity for CNN acceleration on general-purpose processors due to more irregularity in the access pattern and control flow. Also, Winograd-ReLU pruning cannot be applied to conventional CNN models since the new computation in Equation 2 does not correspond to the original convolution operation. It requires changing the network structure and retraining the network from scratch.

## 3 SPATIAL-WINOGRAD PRUNING

In this paper, to achieve a high Winograd-domain weight sparsity on deep CNN models without changing network structures, we propose the *spatial-Winograd pruning*. As shown in Figure 2, it consists of two parts: spatial structured pruning and Winograd direct pruning.

In spatial structured pruning, spatial-domain weights are pruned in a structured way and then retrained to regain the original accuracy. After spatial structured pruning, the weights of the pruned model will be transferred into and kept in the Winograd domain. The Winograd direct pruning then

performs pruning and retraining directly onto the weights in the Winograd domain. The pruning and retraining steps in both spatial structured pruning and Winograd direct pruning will be iteratively executed until we achieve the desired sparsity or the produced model loses much accuracy.

## 3.1 SPATIAL STRUCTURED PRUNING

The first part of the spatial-Winograd pruning is spatial structured pruning. Spatial-domain weights which affect the same Winograd-domain weight are clustered into the same group. Less important weight groups are removed, and the pruned network will be retrained to regain the accuracy. This structured pruning method can help transfer more spatial-domain sparsity into the Winograd domain.

**Spatial Pruning**    For each spatial-domain filter $W$, we need to generate a mask $M^{spatial}$ to indicate the redundant weights. Assuming $W$ has a size of $n \times n$ and the Winograd-domain filter $Q$ is $m \times m$, we have $Q = GWG^\top$. Each element of the Winograd-domain filter, $Q_{i,j}$, is the weighted sum of the spatial-domain weights:

$$Q_{i,j} = \sum_{0 \leqslant u,v \leqslant n-1} (S_{i,j,u,v} \cdot W_{u,v}) \qquad 0 \leqslant i,j \leqslant m-1 \tag{3}$$

where $S$ is a 4D tensor containing the weight coefficients of the spatial-domain weights and is only determined by $m$ and $n$. Details about the calculation of $S$ can be found in Appendix A.1.

For each Winograd-domain weight $Q_{i,j}$, we can create a set $\mathbb{D}_{i,j}$ containing the spatial-domain weights which affect the value of $Q_{i,j}$. $\mathbb{D}_{i,j}$ is defined as

$$\mathbb{D}_{i,j} = \{W_{u,v} \mid S_{i,j,u,v} \neq 0,\ 0 \leqslant u,v \leqslant n-1\} \tag{4}$$

In this case, for each weight group $\mathbb{D}_{i,j}$, we use a function $h(\mathbb{D}_{i,j})$ to measure its importance. In this paper, we use the maximum norm function as $h$

$$h(\mathbb{D}_{i,j}) = \max(\{\ |W_{u,v}| \mid W_{u,v} \in \mathbb{D}_{i,j}\}) \tag{5}$$

With a specific threshold $t^{spatial}$, if $h(\mathbb{D}_{i,j}) < t^{spatial}$, then $\mathbb{D}_{i,j}$ is considered as redundant and all weights included need to be removed. In this case, the corresponding $Q_{i,j}$ will be fixed to 0 and also removed. The set of redundant weights for entire $W$ is the union of all redundant $\mathbb{D}_{i,j}$ and can be calculated as

$$\mathbb{D} = \bigcup_{0 \leqslant i,j \leqslant m-1, h(\mathbb{D}_{i,j}) < t^{spatial}} \mathbb{D}_{i,j} \tag{6}$$

Here we define $\mathbb{D}_{i,j}$ in a structured way based on the relation between spatial-domain weights and Winograd-domain weights. It helps transfer as much spatial-domain sparsity into Winograd-domain sparsity as possible.

The mask matrix $M^{spatial}$ can be generated by

$$M^{spatial}_{u,v} = \begin{cases} 0 & W_{u,v} \in \mathbb{D} \\ 1 & W_{u,v} \notin \mathbb{D} \end{cases} \qquad 0 \leqslant u,v \leqslant n-1 \tag{7}$$

**Spatial Retraining**    After spatial pruning, we can perform the spatial retraining with conventional training algorithms, e.g., stochastic gradient descent (SGD). The removed weights are fixed to 0 by applying $W = W \odot M^{spatial}$ after each training iteration. $\odot$ is element-wise multiplication. The steps of spatial pruning and spatial retraining will be iteratively performed until the retrained model loses much accuracy. The threshold $t^{spatial}$ is gradually increased to incur more sparsity into the network.

In spatial structured pruning, both pruning and retraining steps are performed in the spatial domain. It helps avoid the Winograd-domain retraining to accelerate the pruning process but, at the same time, incurs high Winograd-domain sparsity.

## 3.2 WINOGRAD DIRECT PRUNING

After spatial structured pruning, as in sparse Winograd convolution, weights of the pruned model will be transferred into and kept in the Winograd domain. In Winograd direct pruning, we measure

the importance of each weight based on its impact on output activations, and unimportant weights are removed. The pruned network is then retrained in the Winograd domain, and an importance factor matrix is deployed to adjust the weight gradients.

**Winograd Pruning**  Similar to spatial pruning, in Winograd pruning, we need to generate a mask matrix $M^{Winograd}$ for each Winograd-domain filter $Q$ to indicate the redundant weights. With the weight filter $Q$ in the Winograd domain, the output tile $O$ is calculated as

$$O = A^\top [Q \odot (B^\top I\ B)]A \tag{8}$$

Each output element can be considered as the weighted sum of the products of weights and inputs

$$O_{x,y} = \sum_{0 \leqslant i,j,s,t \leqslant m-1} (H_{x,y,i,j,s,t} \cdot Q_{i,j} \cdot I_{s,t}) \quad\quad 0 \leqslant x,y \leqslant m-n \tag{9}$$

where $H$ is a 6D tensor containing the weight coefficients of different products ($Q_{i,j} \cdot I_{s,t}$) and is only determined by $m$ and $n$. Details about the calculation of $H$ can be found in Appendix A.2.

By removing one weight $Q_{i,j}$, the change on each output $O_{x,y}$ is

$$\Delta O_{x,y}|_{Q_{i,j}} = -1 \cdot \sum_{0 \leqslant s,t \leqslant m-1} (H_{x,y,i,j,s,t} \cdot Q_{i,j} \cdot I_{s,t}) \quad\quad 0 \leqslant x,y \leqslant m-n \tag{10}$$

In Winograd pruning, we need to remove a certain amount of weights while minimizing the change of the output activations $||\Delta O||_2$. Removing an important weight will lead to a larger change in output activations. Therefore, we propose to measure the importance of each weight $Q_{i,j}$ by the expected value of $||\Delta O|_{Q_{i,j}}||_2^2$. In this case, we have

$$
\begin{aligned}
E(||\Delta O|_{Q_{i,j}}||_2^2) &= E\Big( \sum_{0 \leqslant x,y \leqslant m-n} \Big[ \sum_{0 \leqslant s,t \leqslant m-1} (H_{x,y,i,j,s,t} \cdot Q_{i,j} \cdot I_{s,t}) \Big]^2 \Big) \\
&= Q_{i,j}^2 \cdot \Big\{ \sum_{0 \leqslant x,y \leqslant m-n, 0 \leqslant s,t \leqslant m-1} \big[ H_{x,y,i,j,s,t}^2 \cdot E(I_{s,t}^2) \big] + \\
&\quad\quad \sum_{\substack{0 \leqslant x,y \leqslant m-n \\ 0 \leqslant s,t,s',t' \leqslant m-1 \\ (s,t) \neq (s',t')}} \big[ H_{x,y,i,j,s,t} \cdot H_{x,y,i,j,s',t'} \cdot E(I_{s,t} \cdot I_{s',t'}) \big] \Big\}
\end{aligned}
\tag{11}
$$

For simplicity, we can assume input values are independent and identically distributed (i.i.d.), and have expected values of 0. With this assumption, we have

$$E(I_{s,t} \cdot I_{s',t'}) = E(I_{s,t}) \cdot E(I_{s',t'}) = 0 \quad\quad (s,t) \neq (s',t') \tag{12}$$

Since the importance of weights are relative numbers, we can assume $E(I_{s,t}^2) = 1$. In this case,

$$E(||\Delta O|_{Q_{i,j}}||_2^2) = Q_{i,j}^2 \cdot \sum_{0 \leqslant x,y \leqslant m-n, 0 \leqslant s,t \leqslant m-1} H_{x,y,i,j,s,t}^2 \tag{13}$$

Based on Equation. 13, we can generate an importance factor matrix $F$, where

$$F_{i,j} = \sqrt{\frac{E(||\Delta O|_{Q_{i,j}}||_2^2)}{Q_{i,j}^2}} = \sqrt{\sum_{0 \leqslant x,y \leqslant m-n, 0 \leqslant s,t \leqslant m-1} H_{x,y,i,j,s,t}^2} \quad\quad 0 \leqslant i,j \leqslant m-1 \tag{14}$$

Therefore, $F$ is only determined by $m$ and $n$, and keeps the same for all 2D Winograd-domain filters $Q$ in a specific layer. Then Equation. 13 can be simplified to

$$E(||\Delta O|_{Q_{i,j}}||_2^2) = Q_{i,j}^2 \cdot F_{i,j}^2 \tag{15}$$

In this case, with a specific threshold $t^{Winograd}$, we can generate the mask matrix $M^{Winograd}$ as

$$M_{i,j}^{Winograd} = \begin{cases} 0 & Q_{i,j}^2 \cdot F_{i,j}^2 < t^{Winograd} \\ 1 & Q_{i,j}^2 \cdot F_{i,j}^2 \geqslant t^{Winograd} \end{cases} \quad\quad 0 \leqslant i,j \leqslant m-1 \tag{16}$$

For a specific weight $\boldsymbol{Q}_{i,j}$, conventional pruning algorithms (Han et al., 2015b; Guo et al., 2016) use its absolute value $|\boldsymbol{Q}_{i,j}|$ as the weight importance, which is equivalent to using $\boldsymbol{Q}_{i,j}^2$. Therefore, in Equation 16, the employed weight importance, $\boldsymbol{Q}_{i,j}^2 \cdot \boldsymbol{F}_{i,j}^2$, can be considered as using the importance factor matrix $\boldsymbol{F}$ to adjust the conventional weight importance $\boldsymbol{Q}_{i,j}^2$.

**Winograd Retraining**   As the same with the spatial retraining, we fix the removed Winograd-domain weights to 0 by applying $\boldsymbol{Q} = \boldsymbol{Q} \odot \boldsymbol{M}^{Winograd}$ after each training iteration.

However, using conventional SGD to retrain the Winograd-domain parameters will lead to divergence. This is because, as shown in Equation. 14, different locations of Winograd-domain weights have different importance and, therefore, require different learning speeds. Using an extremely small learning rate can avoid the divergence but makes the retraining much slower.

To address this problem, in Winograd retraining, we propose to adjust the gradients of Winograd-domain weights with the importance factor matrix $\boldsymbol{F}$. Assume $loss$ to be the loss value. At the training step $k$, after the backward computation, the gradients of $\boldsymbol{Q}$, $\frac{\partial\,loss}{\partial \boldsymbol{Q}}\big|_k$, will be adjusted by

$$\left(\frac{\partial\,loss}{\partial \boldsymbol{Q}}\bigg|_k\right)^{adjusted} = \frac{\partial\,loss}{\partial \boldsymbol{Q}}\bigg|_k \oslash \boldsymbol{F}^{\circ\alpha} \tag{17}$$

where $\oslash$ and $\circ\alpha$ are the Hadamard division (element-wise division) and Hadamard power (element-wise power of $\alpha$) function, respectively. In this paper, based on empirical results, $\alpha$ is fixed to 1.5. In this case, with the learning rate of $\eta$, the SGD update for the Winograd-domain weights $\boldsymbol{Q}$ at the training step $k$ becomes

$$\boldsymbol{Q}|_{k+1} = \boldsymbol{Q}|_k - \eta \cdot \left(\frac{\partial\,loss}{\partial \boldsymbol{Q}}\bigg|_k \oslash \boldsymbol{F}^{\circ\alpha}\right) \tag{18}$$

## 4   EXPERIMENTS

To evaluate the spatial-Winograd pruning, we perform the experiments on three datasets: CIFAR-10, CIFAR-100 (Krizhevsky, 2009) and ImageNet (ILSVRC-2012) (Russakovsky et al., 2015). Py-Torch (Paszke et al., 2017) is used to implement the pruning framework.

We use the Winograd-ReLU pruning (Liu et al., 2018) as the baseline pruning technique. To show the effectiveness of our proposed method, we test the same models as in Winograd-ReLU pruning: VGG-nagadomi (Nagadomi, 2014), ConvPool-CNN-C (Springenberg et al., 2014) and ResNet-18 (He et al., 2016) on the three datasets tested, respectively. Those models are chosen since the majority of the included convolutional layers use $3 \times 3$ kernels.

For $3\times3$ kernels, we set the input tile size $m$ to 6 instead of 4. A larger input tile size can help achieve higher computation speedup. With our proposed method, we expect that lower input tile sizes can lead to a similar or higher sparsity. This is because, with lower input tile sizes, the spatial-domain weights have less correlation between each other and the spatial structured pruning can achieve a higher sparsity.

### 4.1   CIFAR-10: VGG-NAGADOMI

For the CIFAR-10 dataset, we test the VGG-nagadomi model (Nagadomi, 2014). It contains 8 convolutional layers with $3 \times 3$ kernels. We use batch normalization instead of dropout to regularize the convolutional layers. The original model has a prediction accuracy of 93.96%. We prune the first convolutional layer with a fixed Winograd-domain sparsity of 20%. For the remaining convolutional layers, we incur a uniform Winograd-domain sparsity, increasing from 20% to 80%, for simplicity.

Figure 3a shows the pruning results. The baseline result reported in (Liu et al., 2018) is shown as the dashed line. With $<0.1\%$ accuracy loss, it achieves a sparsity of 60%. With spatial-Winograd pruning, we can achieve a Winograd-domain sparsity of 63%. It is similar to Winograd-ReLU pruning, but spatial-Winograd pruning does not require changing the network structure.

### 4.2   CIFAR-100: CONVPOOL-CNN-C

For the CIFAR-100 dataset, the ConvPool-CNN-C model (Springenberg et al., 2014) is tested. It contains 9 convolutional layers, in which 7 layers use $3 \times 3$ kernels. The original model has a

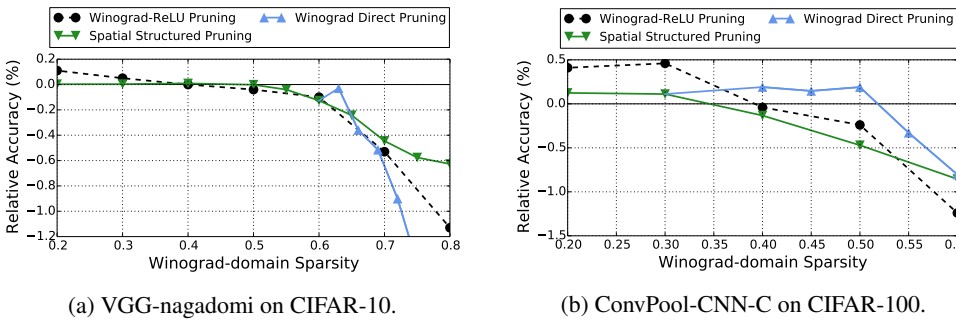

(a) VGG-nagadomi on CIFAR-10.  (b) ConvPool-CNN-C on CIFAR-100.

Figure 3: Pruning of (a) VGG-nagadomi on CIFAR-10 (b) ConvPool-CNN-C on CIFAR-100 with uniform sparsity across layers.

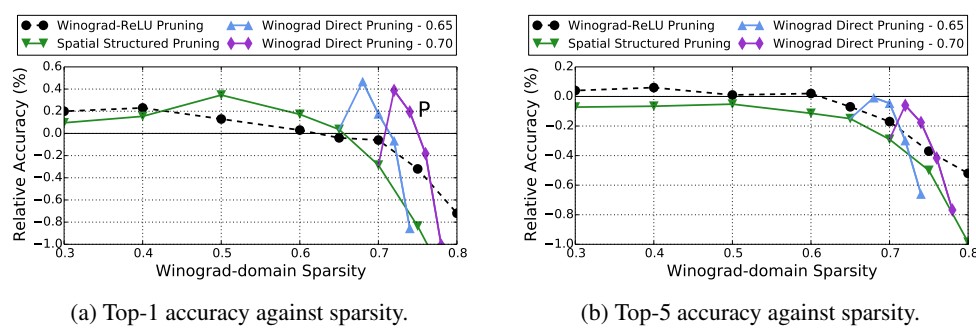

(a) Top-1 accuracy against sparsity.  (b) Top-5 accuracy against sparsity.

Figure 4: Pruning of ResNet-18 on ImageNet with uniform sparsity across the pruned layers.

prediction accuracy of 69.95%. We prune the first convolutional layer with a fixed Winograd-domain sparsity of 20%. Similar to the VGG-nagadomi model, the remaining 6 convolutional layers with $3 \times 3$ kernels are iteratively pruned and retrained with uniform Winograd-domain sparsities.

Figure. 3b shows the result of the relative accuracy against the Winograd-domain sparsity. The baseline result reported in (Liu et al., 2018) is shown as the dashed line. With $<0.1\%$ accuracy loss, it achieves a sparsity of 40%. Winograd direct pruning is applied to the model pruned by spatial structured pruning with 30% sparsity. With no accuracy loss, spatial-Winograd pruning can reach a sparsity of 50%, which is 10% higher than Winograd-ReLU pruning.

## 4.3 IMAGENET: RESNET-18

We test the ResNet-18 model on the ImageNet (ILSVRC-2012) dataset. As the same with Winograd-ReLU pruning, we replace each $2 \times 2$-stride $3 \times 3$ convolutional layer with a $2 \times 2$-stride max-pooling layer followed by a $1 \times 1$-stride $3 \times 3$ convolutional layer. This change makes it easier to apply Winograd convolution on most of the convolutional layers.

The original model has a top-1/top-5 prediction accuracy of 69.82%/89.55%. However, for Winograd-ReLU pruning, Liu et al. (2018) use the model with the original top-1/top-5 accuracy of only 66.67%/87.42%. Despite this, we still use the relative accuracies reported in (Liu et al., 2018) as the baseline.

We prune the 16 convolutional layers in the residual blocks with the same Winograd-domain sparsity. The first convolutional layer and the downsample layers are kept intact. Figure. 4 shows the results of the relative accuracy against the Winograd-domain sparsity. As the dashed line show, the Winograd-ReLU pruning achieves a sparsity of 70%/65% with $<0.1\%$ top-1/top-5 accuracy loss.

We apply Winograd direct pruning to the models pruned by spatial structured pruning with 65% and 70% Winograd-domain sparsity, annotated as Winograd direct pruning - 0.65 and 0.70, respectively. As shown in the figure, with $<0.1\%$ top-1/top-5 accuracy loss, applying Winograd direct pruning to the model with 70% Winograd-domain sparsity can achieve a higher sparsity of 74%/72%. This is

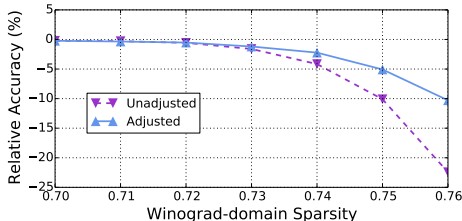 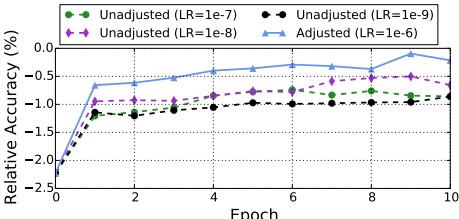

(a) Winograd pruning: unadjusted or adjusted weight importance for different locations.

(b) Winograd retraining: unadjusted or adjusted gradients for different locations..

Figure 5: Effectiveness of employing importance factor matrix $F$ in (a) Winograd pruning and (b) Winograd retraining.

because, with the sparsity increasing, Winograd direct pruning makes the prediction accuracy drop much faster than spatial structured pruning. Although we can use the importance factor matrix $F$ to adjust the weight gradients to accelerate the Winograd-domain retraining, the learning rate still needs to be much lower than in the spatial retraining. In this case, the accuracy loss recovered through Winograd retraining is limited, which makes the accuracy drop much faster when applying Winograd direct pruning.

### 4.4 EFFECTIVENESS OF IMPORTANCE FACTOR MATRIX

In Winograd direct pruning, we use the importance factor matrix $F$ to adjust the weight importance and gradients for different locations of Winograd-domain weights. Here we test the effectiveness of employing the importance factor matrix in both the Winograd pruning and retraining.

We first test how the importance factor matrix $F$ helps in Winograd pruning. Winograd pruning without retraining is applied to the model pruned by spatial structured pruning with 70% sparsity. Figure 5a shows the relative accuracy against the sparsity when pruning with weight importance unadjusted or adjusted with $F$. As shown in the figure, adjusting the weight importance with the importance factor matrix can dramatically reduce the accuracy loss when performing Winograd pruning. When pruning the model to 76% sparsity, using the absolute value as the weight importance will cause a 22% accuracy loss. In comparison, using the importance factor matrix to adjust the weight importance can help reduce the accuracy loss to 10%.

We also test the effectiveness of the importance factor matrix in Winograd retraining. For the model pruned with spatial structured pruning (70% sparsity), Winograd pruning is applied to increase the sparsity to 74%. We then perform Winograd retraining for 10 epochs. Figure 5b shows the relative accuracy against the retraining epochs with unadjusted and adjusted gradients. For unadjusted gradients, we try three learning rates of 1e-7, 1e-8 and 1e-9. Higher learning rates, e.g., 1e-6, will lead to an accuracy drop through retraining. As shown in the figure, adjusting the gradients with the importance factor matrix can substantially accelerate the convergence. With retraining of only 10 epochs, it reduces the accuracy loss to 0.2% while retraining without gradient adjustment only reduces the accuracy loss to 0.7%.

## 5 CONCLUSION

In this paper, we present a new pruning method, spatial-Winograd pruning, to improve the Winograd-domain weight sparsity without changing network structures. It includes two steps: spatial structured pruning and Winograd direct pruning. In spatial structured pruning, we prune the spatial-domain weights based on the internal structure in the Winograd transformation. It can help efficiently transfer the spatial-domain sparsity into the Winograd domain. For Winograd direct pruning, we perform both pruning and retraining in the Winograd domain. An importance factor matrix is proposed to adjust the weight gradients in Winograd retraining, which makes it possible to effectively retrain the Winograd-domain network to regain the original accuracy without changing the network structure. We evaluate spatial-Winograd pruning on three datasets, CIFAR-10, CIFAR-100, ImageNet, and it can achieve the Winograd-domain sparsities of 63%, 50%, and 74%, respectively.

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

# A    COEFFICIENT TENSORS $\boldsymbol{S}$ AND $\boldsymbol{H}$

The coefficient tensors $\boldsymbol{S}$ and $\boldsymbol{H}$ contain the weight coefficients in Equation 3 and Equation 9. They are only determined by the input tile size $m$ and the weight filter size $n$.

## A.1    COEFFICIENT TENSOR $\boldsymbol{S}$

To calculate the tensor $\boldsymbol{S}$, we first introduce an equivalent transformation: for two vectors $\boldsymbol{a}$ and $\boldsymbol{b}$ with a size of $m$, and a matrix $\boldsymbol{C}$ with a size of $m \times m$, we have

$$\boldsymbol{a}^\top \boldsymbol{C} \boldsymbol{b} = \vec{\boldsymbol{1}}^\top \left[ (\boldsymbol{a}\boldsymbol{b}^\top) \odot \boldsymbol{C} \right] \vec{\boldsymbol{1}} \tag{19}$$

where $\vec{\boldsymbol{1}}$ is a vector of size $m$ and all its entries are 1.

For matrix $\boldsymbol{S}$, with the weight transform matrix $\boldsymbol{G}$, we have $\boldsymbol{Q} = \boldsymbol{G}\boldsymbol{W}\boldsymbol{G}^\top$. Each element in Winograd-domain weight filter $\boldsymbol{Q}$ is calculated as

$$
\begin{aligned}
\boldsymbol{Q}_{i,j} &= \boldsymbol{G}_{i,:} \cdot \boldsymbol{W} \cdot (\boldsymbol{G}_{j,:})^\top \\
&= \vec{\boldsymbol{1}}^\top \left[ ((\boldsymbol{G}_{i,:})^\top \boldsymbol{G}_{j,:}) \odot \boldsymbol{W} \right] \vec{\boldsymbol{1}} \\
&= \sum_{0 \leqslant u,v \leqslant n-1} (\boldsymbol{G}_{i,u} \cdot \boldsymbol{G}_{j,v} \cdot \boldsymbol{W}_{u,v})
\end{aligned} \tag{20}
$$

where $0 \leqslant i, j \leqslant m-1$. In this case, compared with Equation 3, each element in the coefficient tensor $\boldsymbol{S}$ can be calculated as

$$\boldsymbol{S}_{i,j,u,v} = \boldsymbol{G}_{i,u} \cdot \boldsymbol{G}_{j,v} \tag{21}$$

where $0 \leqslant i, j \leqslant m-1, 0 \leqslant u, v \leqslant n-1$.

## A.2    COEFFICIENT TENSOR $\boldsymbol{H}$

With the Winograd-domain weight filter $\boldsymbol{Q}$, the output tile $\boldsymbol{O}$ is calculated as

$$\boldsymbol{O} = \boldsymbol{A}^\top [\boldsymbol{Q} \odot (\boldsymbol{B}^\top \boldsymbol{I} \boldsymbol{B})] \boldsymbol{A} \tag{22}$$

Each element $\boldsymbol{O}_{x,y}$ is calculated as

$$\boldsymbol{O}_{x,y} = (\boldsymbol{A}_{:,x})^\top [\boldsymbol{Q} \odot (\boldsymbol{B}^\top \boldsymbol{I} \boldsymbol{B})] \boldsymbol{A}_{:,y} \tag{23}$$

where $0 \leqslant x, y \leqslant m-n$. Based on Equation 19, we have

$$\boldsymbol{O}_{x,y} = \vec{\boldsymbol{1}}^\top [(\boldsymbol{A}_{:,x}(\boldsymbol{A}_{:,y})^\top) \odot \boldsymbol{Q} \odot (\boldsymbol{B}^\top \boldsymbol{I} \boldsymbol{B})] \vec{\boldsymbol{1}} \tag{24}$$

Let $\boldsymbol{V} = (\boldsymbol{A}_{:,x}(\boldsymbol{A}_{:,y})^\top) \odot \boldsymbol{Q} \odot (\boldsymbol{B}^\top \boldsymbol{I} \boldsymbol{B})$, then

$$\boldsymbol{V}_{i,j} = (\boldsymbol{A}_{i,x}\boldsymbol{A}_{j,y}) \cdot \boldsymbol{Q}_{i,j} \cdot (\boldsymbol{B}^\top \boldsymbol{I} \boldsymbol{B})_{i,j} \tag{25}$$

where $0 \leqslant i, j \leqslant m-1$. The element $(\boldsymbol{B}^\top \boldsymbol{I} \boldsymbol{B})_{i,j}$ can be calculated as

$$
\begin{aligned}
(\boldsymbol{B}^\top \boldsymbol{I} \boldsymbol{B})_{i,j} &= (\boldsymbol{B}_{:,i})^\top \boldsymbol{I} \boldsymbol{B}_{:,j} \\
&= \vec{\boldsymbol{1}}^\top [(\boldsymbol{B}_{:,i}(\boldsymbol{B}_{:,j})^\top) \odot \boldsymbol{I}] \vec{\boldsymbol{1}} \\
&= \sum_{0 \leqslant s,t \leqslant m-1} [(\boldsymbol{B}_{s,i}\boldsymbol{B}_{t,j}) \cdot \boldsymbol{I}_{s,t}]
\end{aligned} \tag{26}
$$

Based on Equation 24, 25 and 26, we have

$$
\begin{aligned}
\boldsymbol{O}_{x,y} &= \sum_{0 \leqslant i,j \leqslant m-1} \boldsymbol{V}_{i,j} \\
&= \sum_{0 \leqslant i,j \leqslant m-1} [(\boldsymbol{A}_{i,x}\boldsymbol{A}_{j,y}) \cdot \boldsymbol{Q}_{i,j} \cdot (\boldsymbol{B}^\top \boldsymbol{I} \boldsymbol{B})_{i,j}] \\
&= \sum_{0 \leqslant i,j \leqslant m-1} \left[ (\boldsymbol{A}_{i,x}\boldsymbol{A}_{j,y}) \cdot \boldsymbol{Q}_{i,j} \cdot \sum_{0 \leqslant s,t \leqslant m-1} [(\boldsymbol{B}_{s,i}\boldsymbol{B}_{t,j}) \cdot \boldsymbol{I}_{s,t}] \right] \\
&= \sum_{0 \leqslant i,j,s,t \leqslant m-1} [(\boldsymbol{A}_{i,x}\boldsymbol{A}_{j,y}\boldsymbol{B}_{s,i}\boldsymbol{B}_{t,j}) \cdot \boldsymbol{Q}_{i,j} \cdot \boldsymbol{I}_{s,t}]
\end{aligned} \tag{27}
$$

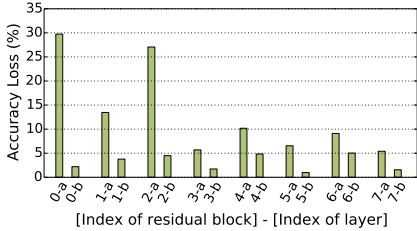

Figure 6: Accuracy loss of ResNet-18 when incurring 60% Winograd-domain sparsity into different layers. Spatial structured pruning is applied with no retraining.

| Layer | Spatial Structured Pruning | | Winograd Direct Pruning |
| | Winograd Sparsity | Corr. Spatial Sparsity | Winograd Sparsity |
|---|---|---|---|
| 0-b | 78.0 % | 88.8 % | 85.0 % |
| 1-b | 77.1 % | 87.9 % | 84.0 % |
| 2-b | 76.4 % | 88.5 % | 84.4 % |
| 3-b | 85.2 % | 93.1 % | 90.4 % |
| 4-b | 76.9 % | 88.9 % | 84.7 % |
| 5-b | 88.6 % | 95.1 % | 93.0 % |
| 6-b | 74.4 % | 88.3 % | 82.4 % |
| 7-b | 82.6 % | 87.8 % | 92.3 % |
| Average | 79.4 % | 88.9 % | 87.6 % |
| Top-1 Acc. | 69.92 % | | 69.94 % |
| Top-5 Acc. | 89.34 % | | 89.51 % |

Table 1: The sparsity for the pruned convolutional layers when pruning the second convolutional layer in each residual block of ResNet-18.

Therefore, compared with Equation 9, each element in the coefficient tensor $\boldsymbol{H}$ is calculated as

$$\boldsymbol{H}_{x,y,i,j,s,t} = \boldsymbol{A}_{i,x}\boldsymbol{A}_{j,y}\boldsymbol{B}_{s,i}\boldsymbol{B}_{t,j} \tag{28}$$

where $0 \leqslant x, y \leqslant m - n, 0 \leqslant i, j, s, t \leqslant m - 1$.

## B  RESNET-18 PRUNING WITH VARIED SPARSITIES ACROSS LAYERS

In addition to pruning ResNet-18 with the same sparsity across all targeting layers, we experiment incurring different sparsities into different layers with spatial-Winograd pruning.

For spatial structured pruning, we first test the pruning sensitivity of each convolutional layer to decide which layers need to be pruned and the corresponding thresholds. To choose the targeting layers, we measure the accuracy loss when 60% of Winograd-domain weights are pruned for each layer. Only one layer is pruned at one time, and other layers are kept intact. Figure. 6 shows the results. In ResNet-18, the $i$-th residual block contains two convolutional layers, $i$-a and $i$-b. As shown in the figure, the first layer in each residual block is much more sensitive to pruning than the second layer. Therefore, we will only prune the second convolutional layer, $i$-b, in each residual block.

For each targeting layer $i$-b, we determine the corresponding pruning threshold $t_i^{spatial}$ based on its pruning sensitivity. We gradually increase the threshold until the validation accuracy drops by 2% and the threshold is recorded as $t_i^{spatial,\,2\%\,loss}$. Then in spatial structured pruning, we can calculate the threshold used for layer $i$-b as $t_i^{spatial} = \beta \cdot t_i^{spatial,\,2\%\,loss}$ where $\beta$ is a multiplier shared across all targeting layers. With a larger $\beta$, the threshold and, therefore, the sparsity will be higher. Also, in Winograd direct pruning, we use the same strategy to choose the thresholds used for different layers.

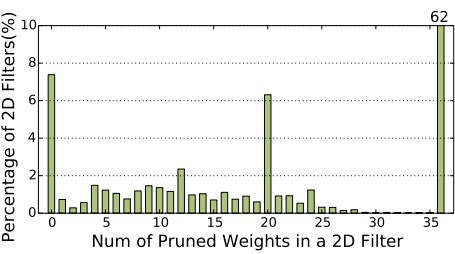

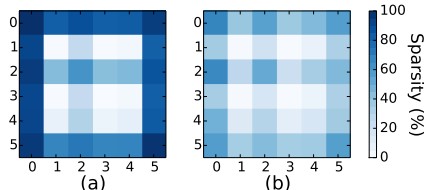

Figure 7: Distribution of 2D filters with different numbers of weights pruned.

Figure 8: Sparsity of different locations of Winograd-domain weights for: (a) filters with 20 weights pruned; (b) filters with at least one weight remaining. Darker locations have higher sparsities.

Table 1 lists the pruning results. After spatial structured pruning, we can reach an average Winograd-domain sparsity of 79.4% for the pruned layers. The corresponding spatial-domain sparsity is 88.9% which is 9.5% higher. Winograd direct pruning can further improve the Winograd-domain sparsity to 87.6% and layer 5-b has the highest sparsity of 93.0%.

## C  SPARSITY DISTRIBUTION

For the pruned ResNet-18 model, we analyze more detailed sparsity distribution across and inside 2D weight filters. Here each Winograd-domain weight matrix $Q$ is considered as a 2D filter. We use the last convolutional layer (7-b) as an example. The model with a uniform sparsity of 74% across all pruned layers, which corresponds to point P in Figure 4a, is tested. Figure 7 shows the sparsity distribution across the filters. As shown in the figure, more than half (62%) of the filters have all weights removed. An interesting observation is that a large portion of the filters have exact 20 weights removed.

To explain why many filters have exact 20 weights removed, we visualize the sparsity distribution inside the filters. Figure 8a shows the sparsity of different locations for the filters with 20 weights removed. Darker locations have higher sparsities where more weights are removed. The border part of the $6 \times 6$ filter, which includes 20 weights, has much higher sparsity than the central part. It means the border part of the Winograd-domain weights is much less important than weights in the central part. A potential reason is that the weights in the central part are correlated to more spatial-domain weights and, therefore, removing them will lead to a larger difference in the output activations. In Figure 8b, we also visualize the sparsity distribution inside the filters with at least one weight remaining, and it shows a similar pattern.

