# OpenReview forum: "Spatial-Winograd Pruning Enabling Sparse Winograd Convolution"
_ICLR.cc/2019/Conference_

### Official Review · AnonReviewer1 · 2018-10-26
**Winograd-aware pruning of convolutions**

**Rating:** 6
**Confidence:** 3

**Review:**

The paper proposes a technique (well, two) to prune convolutional layers to reduce the required amount of computation when  the convolutions are done using the winograd algorithm. Winograd convolutions first transform the image and the filter, apply a multiplication in the transformed space, and then retransform the image back to the intended image space. The transformation of the filter, however, means that sparsity in the regular domain does not translate to sparsity in the winograd domain.

This paper presents two techniques to achieve sparsity in the winograd domain: approximating winograd sparsity based on sparsity in the regular domain (thereby pruning with a non uniform cost model) and pruning in winograd space directly. The actual implementation alternates the first pruning technique and retraining the network with fixed sparsity followed by alternating winograd-space pruning and retraining. The tricky part is retraining in winograd space, which seems to require fine tuned per coordinate learning rates.

My main concern is that the method feels fairly fragile and hyperparameter-heavy: tuning all the learning rates and sparsity rates for all these iterated levels of pruning doesn't seem easy. Similarly, it's unclear why the first stage of pruning is even needed if it's possible to prune and fine tune in winograd space directly. It's unclear from reading the paper how, given a computational budget, to decide the time spent in each phase of the process.

---

> ### Author Response · Authors · 2018-11-14
> **Response to Reviewer 1**
>
> We thank the reviewers for their feedback and valuable time. Please see our response below.
>
> (1) Tuning the learning rate and sparsity?
> We follow these simple rules for choosing the hyperparameters:
>
>         (A) Learning rate:
>                 (a) Spatial structured pruning: Assume for training the original model, the lowest learning rate is β. Then the pruned network is retrained with the learning rates of 10*β and then β.
>                 (b) Winograd direct pruning: For all the retraining process, we use a fixed learning rate of 1e-6.
>
>         (B) Sparsity rate:
>                 (a) Spatial structured pruning: In our evaluation, the sparsity rates used for the iterated levels are mainly based on the settings in the baseline Winograd-ReLU pruning [1]. For practical usage, we can start the pruning with a sparsity of 20%. Then for each iterated level, the sparsity is increased by 10% until the retraining step cannot regain the original accuracy. After this, the sparsity is increased by 5% for every iteration.
>                 (b) Winograd direct pruning: In our evaluation, we manually choose the sparsity rate for each iteration. For practical usage, we can use the same strategy as in spatial structure pruning. In the beginning, for each iterated level, the sparsity is increased by 5% until the retraining step cannot regain the required accuracy. After this, the sparsity is increased by 2%/3% for every iteration.
>
>
> (2) Why the first stage is needed?
> Although we make the Winograd-domain retraining more effective, Winograd direct pruning still makes the prediction accuracy drop much faster than spatial structured pruning. Therefore, applying only Winograd direct pruning cannot achieve the same sparsity level as the current two-step pruning strategy.
>
>
> (3) Time spent in each phase?
> In this paper, we are assuming the computational budget is not limited and focusing on how to improve the Winograd-domain sparsity. We will explore how to distribute a limited computational budget in the future work.
>
> Also, we will try to combine our method with the dynamic network surgery [2] to reduce the computation requirement.  Our current strategy is based on Han et al.’s method [3] in which we generate the pruning mask and then retrain the pruned network. Dynamic network surgery prunes and retrains the network dynamically, which can help reduce the required computation for the pruning process.
>
> ---------------------------------------------------------
> [1] Liu, Xingyu, Jeff Pool, Song Han, and William J. Dally. "Efficient sparse-winograd convolutional neural networks." arXiv preprint arXiv:1802.06367 (2018). https://arxiv.org/abs/1802.06367
> [2] Guo, Yiwen, Anbang Yao, and Yurong Chen. "Dynamic network surgery for efficient dnns." Advances In Neural Information Processing Systems. 2016. https://arxiv.org/abs/1608.04493
> [3] Song Han, Jeff Pool, John Tran, and William J. Dally. “Learning both weights and connections for efficient neural networks.” In NIPS, 2015. https://arxiv.org/abs/1506.02626

---

> > ### Comment · AnonReviewer1 · 2018-11-19
> > **Response to comments**
> >
> > Overall this still seems of limited applicability to me, after reading the reviewer response. Clarifying the details of the learning rates used in the pruning and the tradeoffs involved still makes this seem fairly convoluted and unlikely to generalize to different architectures / models.
> >
> > A more comprehensive discussion of the kinds of speedups (not just sparsity levels but end-to-end speedups) enabled by this and the accuracy tradeoffs we're looking at would help make a future version of this paper more compelling.

---

> > > ### Author Response · Authors · 2018-11-30
> > > **Response for Reviewer's Response**
> > >
> > > We thank the reviewer for the response.
> > >
> > > Considering the generalizability, this is actually what we want to improve compared with the baseline [1]. The baseline solution moves the ReLU function from the spatial domain into the Winograd domain, which fundamentally changes the convolution computation.
> > >
> > > In our solution, we keep the first stage, spatial structured pruning, in the spatial domain since pruning in the spatial domain is well-explored for different network architectures. This stage is as robust as the conventional spatial-domain pruning algorithms. For the next stage of pruning in the Winograd domain, it is difficult to provide a solid mathematical proof. Therefore, we do test different network architectures for different datasets to show that the Winograd direct pruning can be applied to the tested structures.
> > >
> > > For the discussion about the speedups, we agree with the reviewer that a study about the end-to-end speedups is important. We will include more results about real speedups in the future version.
> > >
> > > ----
> > > [1] Liu, Xingyu, Jeff Pool, Song Han, and William J. Dally. "Efficient sparse-winograd convolutional neural networks." arXiv preprint arXiv:1802.06367 (2018). https://arxiv.org/abs/1802.06367

---

### Official Review · AnonReviewer2 · 2018-10-29

**Rating:** 4
**Confidence:** 3

**Review:**

Review of Spatial-Winograd Pruning Enabling Sparse Winograd Convolution

Summary:
In this submission, the authors propose a new method for pruning weights in the presence in CNNs in which the convolution is expressed as a CNN. The goal of the project is to demonstrate that they can achieve a network which contains fewer weights (and runs faster) then the original network with minimal sacrifice in network performance and without altering the network architecture.

Major Comments:
My largest comments and concerns revolve around the degree to which the proposed pruning methods results in a model that is applicable for real world devices.

1. At the minimum, the authors should provide a table with the number of parameters in the (a) original networks, (b) network with baseline pruning method and (c) network with their pruning methods. (Are the savings entirely in convolutional filters or are there savings from fully connected layers?)

2. If the authors are indeed arguing that a goal of network pruning is to perform faster inference, then results must be shown to justify this -- as it is not obvious that speed-ups could be achieved by just pruning weights. In the case of other methods, speed-ups may be achieved by selectively pruning channel filters (as opposed to spatial positions in the convolutional filter) or pruning fully connected layers (in fact, for VGG and AlexNet, a majority of the reduction in parameters due to pruning were found in these layers).

3. Are these levels of sparsity useable in a real-world system? Often the degree of sparsity in "vanilla" CNNs are at levels (e.g. in AlexNet, 30-50% sparsity in higher layers) not high enough to be harnessed for a fast implementation. Selectively zero-ing out individual spatial components of a filter might reduce the parameter count but not achieve any speed up gains. That is, the sparsity must be structured in such a way as to permit a faster implementation [e.g. 1]

4. Considering that one of the baselines changes the network architecture itself [2], I would be curious to understand how effective this method is versus other, simple baselines that change the network architecture such as (1) decreasing the number of filter, (2) decreasing the kernel sizes, (3) swapping a convolutional filter for a separable convolution [3, 4]. All of these baselines are simple to train and experiment with and a practitioner would probably consider in many situations where speed or # parameters are a constraint.

Minor Comments:

- It is unclear from the presentation whether both proposed pruning methods may be trained in tandem or in series. Please clarify in the manuscript.

- Why does Figure 3a, 3b focus on maintaining 20% sparsity on the 1st layer of the network systematically all other layers in sparsity? What is special about the first layer?

- Why do the authors explore a different range of sparsities in Figure 3a and Figure 3b?

- The authors should discuss the source of the variability (and non-monotonicity) in the plots in Figure 3 and 4 for their proposed method. How are we to interpret this? Naively, it would be appear that the method is somewhat unpredictable in performance across a range of sparsity.

- Why do the blue and purple curves in Figure 4 not space the entire range of sparsities?

- Figure 5b. What is the relative accuracy measured with respect to? The baseline model at that particular epoch or at final asymptotic performance?

[1] Outrageously Large Neural Networks: The Sparsely-Gated Mixture-of-Experts Layer
https://openreview.net/forum?id=B1ckMDqlg

[2] Efficient sparse-winograd convolutional neural networks.
Xingyu Liu, Jeff Pool, Song Han, and William J Dally.

[3] MobileNets: Efficient Convolutional Neural Networks for Mobile Vision Applications
Andrew G. Howard, Menglong Zhu, Bo Chen, Dmitry Kalenichenko, Weijun Wang, Tobias Weyand, Marco Andreetto, Hartwig Adam
https://arxiv.org/abs/1704.04861

[4] Xception: Deep Learning with Depthwise Separable Convolutions
François Chollet
https://arxiv.org/abs/1610.02357

---

> ### Author Response · Authors · 2018-11-14
> **Response to Reviewer 2 for the Minor Comments (Part 2/2)**
>
> Response for the minor comments:
>
> (1) Whether trained in tandem or in series?
> We do not start Winograd-domain pruning until we finish spatial structured pruning because the retrained Winograd-domain weights cannot be transformed back into the spatial domain. In both pruning techniques, each pruning+retraining iteration is working on the pruned model generated by the previous iteration.
>
> (2) 20% sparsity in the first layer?
> To make a fair comparison with the baseline [2], we use the same setting that prunes the first layer with a sparsity of 20% (density of 100%-20%=80%). For conventional DNN models, the first layer has much less redundancy than other layers. Without an accuracy loss, we cannot achieve a high sparsity in the first layer. Therefore, for the goal of speedup, we usually do not prune the first layer.
>
> (3) Different range of sparsities in Figure 3a and Figure 3b?
> All figures in Figure 3 and Figure 4 show the results until we lose around 1.0% accuracy. Since only <0.1% accuracy loss is allowed for the experiments, we do not show results for the sparsities beyond 60% in Figure 3b.
>
> (4) The variability of sparsity?
> We hope to answer this question thoroughly. There are two variabilities/non-monotonicity in the plots:
>     A. For either spatial structured pruning or Winograd direct pruning, the accuracy may first go up and then go down. This is because, in the beginning, pruning can reduce overfitting and help achieve higher accuracy. Then with more weights get pruned, the network capacity is reduced and the accuracy goes down. This tendency has been explored in the previous work [7].
>     B. For Winograd direct pruning, we may achieve higher accuracy than the spatial-domain model. This is because the Winograd transformation will increase the number of weight parameters. Also, when directly retraining the Winograd-domain network, we do not need to keep the one-to-one correspondence between the spatial-domain and Winograd-domain weight filters. Each Winograd-domain weight can be trained independently. Therefore, the retrained Winograd-domain network has a higher capability than the spatial-domain network and can achieve higher accuracy.
>
> We agree with the reviewer that the performance (prediction accuracy) across a range of sparsity is unpredictable. However, this is a common problem for the pruning techniques. Due to the lack of mathematical explanations of DNN models, it is difficult to predict the achievable accuracy with a certain sparsity.
>
> (5) Blue and purple curves in Figure 4 not spacing the entire range of sparsities?
> As discussed in response (3) for the minor comments, in the evaluation, we only allow a <0.1% accuracy loss. The endpoints of the blue and purple curves already have a much higher accuracy loss.
>
> (6) What is the relative accuracy measured with respect to?
> The relative accuracy is measured with respect to the baseline accuracy of the original unpruned model (top-1/top-5 prediction accuracy of 69.82%/89.55%).
>
> ---------------------------------------------------------
> [1] Li, Sheng, Jongsoo Park, and Ping Tak Peter Tang. "Enabling sparse Winograd convolution by native pruning." arXiv preprint arXiv:1702.08597 (2017). https://arxiv.org/abs/1702.08597
> [2] Liu, Xingyu, Jeff Pool, Song Han, and William J. Dally. "Efficient sparse-winograd convolutional neural networks." arXiv preprint arXiv:1802.06367 (2018).  https://arxiv.org/abs/1802.06367
> [3] Howard, Andrew G., Menglong Zhu, Bo Chen, Dmitry Kalenichenko, Weijun Wang, Tobias Weyand, Marco Andreetto, and Hartwig Adam. "Mobilenets: Efficient convolutional neural networks for mobile vision applications." arXiv preprint arXiv:1704.04861 (2017). https://arxiv.org/abs/1704.04861
> [4] Zhang, Xiangyu, Xinyu Zhou, Mengxiao Lin, and Jian Sun. "ShuffleNet: An Extremely Efficient Convolutional Neural Network for Mobile Devices." https://arxiv.org/abs/1707.01083
> [5] Sandler, Mark, Andrew Howard, Menglong Zhu, Andrey Zhmoginov, and Liang-Chieh Chen. "Inverted residuals and linear bottlenecks: Mobile networks for classification, detection and segmentation." arXiv preprint arXiv:1801.04381 (2018). https://arxiv.org/abs/1801.04381
> [6] Chollet, François. "Xception: Deep learning with depthwise separable convolutions." arXiv preprint (2017): 1610-02357. https://arxiv.org/abs/1610.02357
> [7] Song Han, Jeff Pool, John Tran, and William J. Dally. “Learning both weights and connections for efficient neural networks.” In NIPS, 2015. https://arxiv.org/abs/1506.02626

---

> ### Author Response · Authors · 2018-11-14
> **Response to Reviewer 2 for the Major Comments (Part 1/2)**
>
> We thank the reviewers for their feedback and valuable time. Please see our response below.
>
> Response for the major comments:
>
> (1) The number of parameters?
> As the reviewer comments, the goal of our technique is for faster inferences. Since Winograd convolution only applies to convolutional layers, all the saving shown in this paper is from the convolutional layers.
>
>
> (2/3) Faster inference in a real-world system?
> The goal of our technique is for faster inferences.
>
> For the layers with 3x3 kernels in AlexNet, the previous sparse Winograd convolution technique [1] can help achieve an up to 5.4x speedup compared with the spatial convolution on Intel Xeon processors. The baseline used in our paper, Winograd-ReLU pruning [2], can already achieve a higher sparsity than the sparse Winograd convolution. Therefore, we expect our technique to achieve a similar or higher speedup.
>
> In networks like ResNet-18, our proposed method can achieve a sparsity of around 70%. As the reviewer comments, it is difficult to utilize this level of sparsity for speedup. Therefore, in Appendix B, we try only pruning the second convolutional layer in each residual block (8 convolutional layers in total). As the results show, we can achieve an average sparsity of 87.6% across the pruned layers, which reaches a level of usable sparsity. We are currently working on implementing the sparse Winograd convolution kernel. In the future work, we will also try pruning channel filters for the other unpruned layers to test the highest speedup we can get for the entire network.
>
> Our work focuses on Winograd convolution. Therefore, as the same with the baseline [2], we are not pruning fully-connected layers but only the convolutional layers. Other pruning techniques applicable to fully-connected layers can be combined with our technique for further acceleration.
>
>
> (4) Comparison with methods changing the network architecture.
> Directly decreasing the number of filters or the kernel sizes usually leads to an accuracy drop while our technique can help accelerate the computation and maintain the same level of accuracy.
>
> In the following table, we list the numbers of the required parameters and computation for various light-weight designs, Xception and also our pruned ResNet-18 model (74% Winograd-domain sparsity) for the ImageNet challenge.
>
> ==============================================
> Networks                             | Top1   | Params |  MAdds
> -------------------------------------------------------------------------
> MobileNetV1 [3]                  |  70.6   |   4.2M    |   575M
> ShuffleNet [4] (1.5*)           |  71.5   |   3.4M    |   292M
> ShuffleNet (x2)                    |  73.7   |   5.4M    |   524M
> MobileNetV2 [5] (0.7)         |  69.8   |   2.6M    |   209M
> MobileNetV2                        |  72.0   |   3.4M    |   300M
> MobileNetV2 (1.4)               |  74.7   |   6.9M    |   585M
> Xception [6]                          |  79.0   |  22.8M   |  8484M
> -------------------------------------------------------------------------
> Pruned ResNet-18 (ours)   | 70.0    |  11.7M   |  262M
> ==============================================
> * For the networks, inside the parentheses is the multiplier for the layer size
>
> For our pruned ResNet-18 model, the remaining parameters and MAdds (multiply-add) are comparable with those light-weight designs. One thing needs to be mentioned is that the first convolutional layer (which is not pruned) in our ResNet-18 model costs 118M MAdds which is 118M/262M = 45% of the total computation.
>
> However, our pruned model performs sparse computation which is less efficient than the conventional dense computation. Therefore, in the future work, we will work on implementing an efficient sparse Winograd convolution library. Also, it is promising to investigate combining the light-weight designs and our proposed spatial-Winograd pruning technique. This combination may help further reduce the computation required for achieving a certain accuracy.
>
> ---------------------------------------------------------
> For the references, please check the next part.

---

> > ### Comment · AnonReviewer2 · 2018-11-26
> > **Response to rebuttal**
> >
> > I have read all of the other reviewer comments as well as the author responses to my original comments.
> >
> > I thank the authors for their thoughtful responses but my review remains at 4: Ok but not good enough - rejection. The primary reasons that review remains at this level are:
> >
> > (1) Practical considerations / real world
> >
> > The authors argue that they can achieve a higher level of sparsity and provide some evidence for this in Appendix B. First, I would suggest to the authors that this should be placed in the main body of the text since this seems a central result. That said, it is still not clear if their gains are realizable. Currently, the authors argue that this will lead to faster inference -- however, this is a theoretical argument and I have seen many methods fail to achieve their theoretical gains when actually implemented. Given that the faster inference is the central contribution of this paper, I strongly believe this requites a proof of concept implementation.
> >
> > (2) Generality of the result
> >
> > As mentioned by other reviewers, the methods appear fragile and far from guaranteed from providing gains across a broad array of architectures. Rather then focus on 1 architecture (e.g. their custom ResNet-18), I would prefer to see them 'swap in' this method across a broad array of architectures (e.g. In the rebuttal table, the authors should focus on MobileNetV1 vs MobileNetV1-Pruned; XCeption vs XCeptio-Pruned, etc.). This would provide more guarantees that in spite of the fragility of the training method, this method is broadly applicable. Note that this is precisely why decreasing filter bank sizes, swapping in separable convolutions, etc. are considered simple but general methods for speeding up networks with minimal harm in prediction accuracy.

---

> > > ### Author Response · Authors · 2018-11-30
> > > **Response for Reviewer's Response**
> > >
> > > We thank the reviewer for the response.
> > >
> > > (1) For the practical considerations, we agree with the reviewer that real speedups verified on hardware platforms are important. We will include more related results in the future version.
> > >
> > > (2) For the generality of the result, this is actually what we want to improve compared with the baseline [1]. The baseline solution moves the ReLU function from the spatial domain into the Winograd domain, which fundamentally changes the convolution computation.
> > >
> > > In our solution, we keep the first stage, spatial structured pruning, in the spatial domain since pruning in the spatial domain is well-explored for different network architectures. This stage is as robust as the conventional spatial-domain pruning algorithms. For the next stage of pruning in the Winograd domain, it is difficult to provide a solid mathematical proof. Therefore, we do test different network architectures for different datasets to show that the Winograd direct pruning can be applied to the tested structures.
> > >
> > > We agree with the reviewer that our solution cannot guarantee a performance gain for every single layer of different DNN architectures compared with the conventional spatial pruning algorithms or the baseline dense model. The performance gain/loss we can have varies for different layers in different networks. Therefore, our future work will explore how to determine which layers can benefit more from the proposed spatial-Winograd pruning compared with conventional network compression techniques.
> > >
> > > As for networks with separable convolutions, e.g., MobileNet, those models are not directly the targets for our proposed method. With separable convolutions, the main part of the parameters and computation is for the 1x1 pointwise convolution. 1x1 convolution is not the target of Winograd convolution and, therefore, not the target of our proposed method. Considering that the networks with separable convolutions have traded accuracy for computation efficiency, a potential method for applying our solution is first converting the 3x3 depthwise convolution back into a full 3x3 convolution to increase the accuracy and then applying our technique. This might help achieve a higher accuracy while maintaining a similar computation/parameters requirement.
> > >
> > > ----
> > > [1] Liu, Xingyu, Jeff Pool, Song Han, and William J. Dally. "Efficient sparse-winograd convolutional neural networks." arXiv preprint arXiv:1802.06367 (2018). https://arxiv.org/abs/1802.06367

---

### Official Review · AnonReviewer3 · 2018-11-06
**spatial-Winograd pruning**

**Rating:** 5
**Confidence:** 3

**Review:**

In this paper, the authors propose a spatial-Winograd pruning framework, consisting of spatial structured pruning and Winograd direct pruning.  First, spatial structured pruning allows the pruned weight in the spatial domain to be kept in the Winograd domain. Then, Winograd direct pruning can further improve the sparsity in the Winograd domain. The organization of paper is OK.  The main concern is about the practical part. In the experiment, the advantage of propose framework is marginal, compared to the relevant approaches.  More comparisons with the state-of-the-art approaches should be investigated, such as light-weight design (MobileNet, ShuffleNet).

---

> ### Author Response · Authors · 2018-11-14
> **Response to Reviewer 3**
>
> We thank the reviewers for their feedback and valuable time. Please see our response below.
>
>
> (1) Marginal improvements?
> Although our proposed method does not achieve a much higher sparsity than the baseline [1], the expected performance improvement is considerable. With the sparse Winograd convolution, the execution time of the convolutional layer is proportional to the density of the weights. Therefore, as an example, with <0.1% top-1 accuracy loss on the ResNet-18 model, our technique can increase the sparsity from 70% to 74%. The expected execution time reduction is 100% - (100%-74%)/(100%-70%) = 13%.
>
> Also, our proposed technique can help avoid changing the network structure.
>
> In the future work, for the baseline Winograd-ReLU pruning [1], we will test whether adjusting the gradients with our proposed importance factor matrix can help further increase the Winograd-domain sparsity on certain networks.
>
>
> (2) Comparison with the light-weight designs?
> In the following table, we list the numbers of the required parameters and computation for various light-weight designs, Xception and also our pruned ResNet-18 model (74% Winograd-domain sparsity) for the ImageNet challenge.
>
> ==============================================
> Networks                             | Top1   | Params |  MAdds
> -------------------------------------------------------------------------
> MobileNetV1 [2]                  |  70.6   |   4.2M    |   575M
> ShuffleNet [3] (1.5*)           |  71.5   |   3.4M    |   292M
> ShuffleNet (x2)                    |  73.7   |   5.4M    |   524M
> MobileNetV2 [4] (0.7)         |  69.8   |   2.6M    |   209M
> MobileNetV2                        |  72.0   |   3.4M    |   300M
> MobileNetV2 (1.4)               |  74.7   |   6.9M    |   585M
> Xception [5]                          |  79.0   |  22.8M   |  8484M
> -------------------------------------------------------------------------
> Pruned ResNet-18 (ours)   | 70.0    |  11.7M   |  262M
> ==============================================
> * For the networks, inside the parentheses is the multiplier for layer size
>
> For our pruned ResNet-18 model, the remaining parameters and MAdds (multiply-add) are comparable with those light-weight designs. One thing needs to be mentioned is that the first convolutional layer (which is not pruned) in our ResNet-18 model costs 118M MAdds which is 118M/262M = 45% of the total computation.
>
> However, our pruned model performs sparse computation which is less efficient than the conventional dense computation. Therefore, in the future work, we will work on implementing an efficient sparse Winograd convolution library. Also, it is promising to investigate combining the light-weight designs and our proposed spatial-Winograd pruning technique. This combination may help further reduce the computation required for achieving a certain accuracy.
>
> ---------------------------------------------------------
> [1] Liu, Xingyu, Jeff Pool, Song Han, and William J. Dally. "Efficient sparse-winograd convolutional neural networks." arXiv preprint arXiv:1802.06367 (2018).   https://arxiv.org/abs/1802.06367
> [2] Howard, Andrew G., Menglong Zhu, Bo Chen, Dmitry Kalenichenko, Weijun Wang, Tobias Weyand, Marco Andreetto, and Hartwig Adam. "Mobilenets: Efficient convolutional neural networks for mobile vision applications." arXiv preprint arXiv:1704.04861 (2017). https://arxiv.org/abs/1704.04861
> [3] Zhang, Xiangyu, Xinyu Zhou, Mengxiao Lin, and Jian Sun. "ShuffleNet: An Extremely Efficient Convolutional Neural Network for Mobile Devices." https://arxiv.org/abs/1707.01083
> [4] Sandler, Mark, Andrew Howard, Menglong Zhu, Andrey Zhmoginov, and Liang-Chieh Chen. "Inverted residuals and linear bottlenecks: Mobile networks for classification, detection and segmentation." arXiv preprint arXiv:1801.04381 (2018). https://arxiv.org/abs/1801.04381
> [5] Chollet, François. "Xception: Deep learning with depthwise separable convolutions." arXiv preprint (2017): 1610-02357. https://arxiv.org/abs/1610.02357

---

### Meta-Review · Area_Chair1 · 2018-12-02
**Reject**

**Confidence:** 4
**Recommendation:** Reject

**Metareview:**

Reviewer scores straddle the decision boundary but overall this does work does not meet the bar yet. Even after discussion with the authors, the reviewers reconfirmed there 'reject' recommendation and the area chair agrees with that assessment.